# Numerical Simulation of Electrical Properties for Pore-Scale Hydrate-Bearing Sediments with Different Occurrence Patterns and Distribution Morphologies

Xixi Lan [1,2], Changchun Zou [1,2,*] , Cheng Peng [1,2,*] , Caowei Wu [1,2] , Yuanyuan Zhang [1,2] and Shengyi Wang [1,2]

[1] School of Geophysics and Information Technology, China University of Geosciences, Beijing 100083, China; lxx_cugb@163.com (X.L.)
[2] National Engineering Research Center of Offshore Oil and Gas Exploration, Beijing 100028, China
* Correspondence: zoucc@cugb.edu.cn (C.Z.); pengc@cugb.edu.cn (C.P.)

**Abstract:** Characterizing the electrical properties of hydrate-bearing sediments, especially resistivity, is essential for reservoir identification and saturation evaluation. The variation in electrical properties depends on the evolution of pore habits, which in turn are influenced by the hydrate growth pattern. To analyze the relationship between hydrate morphology and resistivity quantitatively, different micromorphologies of hydrates were simulated at the pore scale. This study was also conducted based on Maxwell's equations for a constant current field. During numerical simulation, three types of hydrate occurrence patterns (grain-cementing, pore-filling and load-bearing) and five types of distribution morphologies (circle, square, square rotated by 45°, ellipse and ellipse rotated by 90°) in the pore-filling mode were considered. Moreover, the effects of porosity, the conductivity of seawater, the size of the pore-throat and other factors on resistivity are also discussed. The results show that the variation in resistivity with hydrate saturation can be broadly divided into three stages (basically no effect, slow change and rapid growth). Compared with the grain-cementing and pore-filling modes, the resistivity of the load-bearing mode was relatively high even when hydrate saturation was low. For high hydrate-saturated sediments ($S_h > 0.4$), the saturation exponent $n$ in Archie equation was taken as $2.42 \pm 0.2$. The size of the throat is furthermore the most critical factor affecting resistivity. This work shows the potential application prospects of the fine reservoir characterization and evaluation of hydrate-bearing sediments.

**Keywords:** hydrate-bearing sediment; micromorphology in porous media; electrical property; numerical modeling; saturation exponent $n$

## 1. Introduction

In recent years, the energy demand in the world has increased while the reserves of fossil fuels have been shrinking. Therefore, it is urgent to find new energy sources to alleviate the problem of energy shortage in the future. Natural gas hydrate (NGH) is an ice-like crystalline solid compound formed by water molecules and other guest molecules such as methane, ethanol carbon dioxide and so on, under certain temperature and pressure conditions [1]. It is widely distributed in submarine sediments on continental shelf margins and in permafrost regions [2,3]. Due to its wide distribution, huge resources and high density (1 m³ of NGH can release 164~180 m³ methane gas and its energy density is nearly 10 times that of traditional energy), NGH is regarded as a new potential source with cleanliness, high efficiency and abundant reserves [4–7].

Since NGH is stable only under low-temperature and high-pressure conditions, it will decompose and transform from a solid to gas state once conditions change. Natural gas spills into sediment easily, leading to formation deficits and causing large uncontrollable phase changes. Moreover, methane in natural gas hydrate is a gas with a strong greenhouse effect. Instability of hydrate reservoirs may cause submarine landslide, borehole collapse

and methane gas leakage, which may lead to tsunamis, the greenhouse effect and other disasters and bring great inconvenience to exploration and development, so there is an immediate need to carry out research on the fine reservoir characterization and evaluation of marine NGH [8–10]. Resistivity is very sensitive to the existence of NGH. In the rock skeleton of sedimentary rocks, only the solid particles of rock minerals contain a small number of free electrons, resulting in a poor electrical conductivity of the rock skeleton, which mainly relies on the ionic conductivity of fluids between the connected pores and fracture. When NGH forms in the pore space of seafloor sediments, the flow channels are blocked, leading to reduced porosity and the geophysical response is characterized by high resistivity. Consequently, the anomalous change in resistivity can be used to indicate the hydrate-bearing reservoir and also to estimate the hydrate saturation to a certain extent [11–13].

Based on previous research, it is evident that the factors affecting the electrical properties of hydrate-bearing reservoirs mainly include the temperature and pressure of the reservoir, salinity of pore water, hydrate-bearing saturation, structure of the pore space and clay content of the reservoir [14,15]. Temperature on the one hand can affect the solubility of charged ions, such that the concentration of ions in the pore solution changes; on the other hand, it can also affect the migration rate of ions [16–18]. Seafloor sediments are mostly unconsolidated porous media, and pressure mainly affects the porosity of their porous media. When the pore fluid is a salt solution, the presence of salt ions can greatly enhance electrical conductivity, causing a significant decrease in the resistivity value of the sediment [19]. The hydrate will block the throats and pores in the formation, leading to a poor pore connectivity. Meanwhile, the fluid may be trapped in a closed range, and the ions in the fluids will not be able to communicate well. Thus, the electrical conductivity will be poor [20]. The pore structure of the reservoir refers to the size, shape, distribution and connectivity of the throats and pores, which all have a certain influence on the electrical properties of the rock [21]. When clay is present in the reservoir, the electrical properties become complex and the double layer effect caused by it produces additional conductivity [22,23]. In addition, it has been shown in the literature that the electrical conduction patterns of sediments differ under different hydrate occurrence conditions. This is mainly attributable to the change in pore structure caused by the growth of hydrate, which in turn affects electrical properties [24–26]. However, most of the aforementioned studies are qualitative analyses, which basically analyze the general trend of the influence of these factors on resistivity, and further work can be completed on the quantitative analysis of resistivity changes. It is worth noting that multiple phases of solid (hydrate, ice), liquid (seawater) and gas (free gas) may co-exist in hydrate-bearing sediments, making current conduction much more complex. In general, both solid and gas phases are considered to be insulating and cause an increase in resistivity in electrical properties, especially for ice and hydrates, both of which can block the pore and throat. It is currently difficult to distinguish them by their resistivity information alone [27,28]. At present, the study of hydrate electrical properties mainly focuses on the study of sediment pore space and the double layer effect brought by and clay, though the double layer effect is not particularly significant in seawater with high salinity. The electrical conduction in the pore space of hydrate-bearing sediments is extremely complex and its mechanism is not completely clear, which brings a challenge to quantitative analysis. To correctly understand electrical conduction characteristics, it is necessary to consider hydrate occurrence patterns and electrical conduction channels at the pore scale.

Currently, the technology to obtain in situ NGH samples is immature and extremely costly. It is almost impossible to achieve absolute preservation of NGH under in situ conditions [29]. Laboratory simulation is an important method for studying the evolutionary law of NGH's physical properties. Moreover, it can also provide support for reservoir understanding during hydrate exploration and development in the field [30–33]. Laboratory simulation for hydrates usually includes core experiments and numerical simulations of the petrophysical type. The core experiments conducted in the laboratory are the basis for

the research on the electrical conductivity mechanism of hydrates, which allows visual observation of the effects of temperature, pressure, sediment salinity and other factors on resistivity [34–37]. However, they are often unable to accurately analyze the extent of influence of microscopic factors such as rock components and pore structure on electrical properties. Based on the construction of numerical rock physics models, a conductivity model can study the microscopic conductivity mechanism of porous media and quantitatively obtain the relationship between the pore structure as well as fluid distribution and rock electrical properties [38,39]. Numerical simulations of the petrophysical variety can compensate for the above-mentioned laboratory deficiencies and are not limited by the hydrate formation environment and related experimental conditions, which is well suited for the study of hydrates' electrical properties at the pore scale.

This paper focuses on the electrical properties of hydrate-bearing sediments at the pore scale, which were mainly simulated using the finite element method. This study was also conducted based on Maxwell's equations for a constant current field. The effects of two arrangements with hydrate skeleton grains, three occurrence patterns and five distribution morphologies on hydrate-bearing sediments were investigated mainly by simulating different variations of hydrate saturation. Additionally, the relationship between the resistivity index and water saturation ($RI$–$S_w$) was fitted to the Archie formula to obtain the saturation exponent $n$, and the $n$ of this paper is compared with previous work. Finally, the influences of the skeleton grain arrangements of the sediment, initial porosity, hydrate saturation, conductivity of seawater, tortuosity and size of the pore-throat on resistivity are also discussed.

## 2. Method

### 2.1. The Occurrence Patterns of Hydrates at the Microscopic Scale

The occurrence pattern and distribution morphology of hydrates in sediments is the contact relationship between hydrate and sediment skeleton grains and their position in a pore space, which describes the formation of hydrate at the microscopic scale. To determine the microscopic morphology of pore-type gas hydrate reservoirs in a submarine, many researchers have carried out a series of hydrate synthesis–decomposition experiments and observed the microscopic distribution of hydrates using XCT, SEM and other methods. The results show that hydrates can form and reach different occurrence patterns and distribution morphologies under different gas source and sediment composition conditions, which is worthwhile to study further [40–42].

From the perspective of the pore scale, the occurrence patterns of hydrates in sediments can roughly be divided into three main types: (a) grain-cementing (GC) mode, where hydrate grows uniformly on and around the surface of skeleton grains and eventually wraps around and cements the skeleton grains; (b) pore-filling (PF) mode, where hydrate is dispersed in the pores of the rock without contacting any sediment skeleton grains; and (c) load-bearing (LB) mode, where hydrate links adjacent skeleton grains together and forms points of contact with them, becoming a part of the skeleton while providing stability to the sediment [43–46]. A schematic diagram of each hydrate mode is shown in Figure 1, which assisted us in visually identifying hydrate occurrence patterns. It is clear that the occurrence pattern and distribution morphology of hydrates in sediment grains have different effects on the electrical properties of the reservoir.

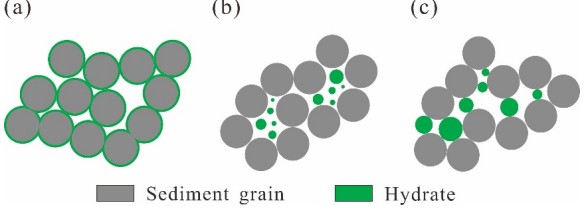

**Figure 1.** The occurrence patterns of hydrates at the microscopic scale: (**a**) grain-cementing (GC) mode; (**b**) pore-filling (PF) mode; and (**c**) load-bearing (LB) mode.

## 2.2. Microscopic Numerical Simulation Methods

A constant current field is generated by a current that does not vary with time. In general, the current intensity does not effectively describe the distribution of the current in the current field. To this end, current density $J$ (A/m$^2$) is introduced to quantitatively describe the current distribution and direction at each point in space. The differential form of the current continuity equation is as follows:

$$\nabla \cdot J + \frac{\partial \rho}{\partial t} = 0 \tag{1}$$

$$\nabla \cdot J = -\frac{\partial \rho}{\partial t} \tag{2}$$

where $\rho$ is the space charge density (C/m) and $t$ is the time of applying current (s).

According to the law of charge conservation, the charge $q$ (C) flowing out from the surface $S$ in unit time should be equal to the reduction in charge from the surface $S$:

$$\iint\limits_{S} J \cdot dS = -\frac{\partial q}{\partial t} \tag{3}$$

In a constant current field, although the charge is in motion, the current does not change with time. Moreover, the distribution of moving charges in space does not change with time, and there can be no increase or decrease in charge in any closed surface $S$, that is, $\frac{\partial q}{\partial t} = 0$. So, Formula (3) is equal to the following:

$$\iint\limits_{S} J \cdot dS = 0 \tag{4}$$

The differential form of the continuity equation for a constant current can be obtained by the Gaussian divergence theorem:

$$\nabla \cdot J = 0 \tag{5}$$

This equation is mathematically similar to the electrostatic equation in free space. In addition, as in the case of static electricity, Maxwell's equations imply that the electric field needs to meet the additional requirements of a non-rotational field [47]:

$$\nabla \times E = 0 \tag{6}$$

where $E$ is the electric field intensity (V/m).

The flow chart of the numerical simulation process based on Maxwell's equations is shown in Figure 2. First, the variables and formulas of the simulation process need to be defined. Then, the two-dimensional (2D) geometric models are constructed, the boundary conditions are set and the mesh is dissected. When the work above is ready, Maxwell's equations can be solved and finally, electrical parameters can be calculated based on general physics concepts.

## 2.3. Simulation of Saturation Model

Generally, after calculating the hydrate-bearing reservoir resistivity of different occurrence patterns by the numerical simulation method, the relation between the formation factor and porosity ($F$–$\varphi$) and the resistivity index and water saturation ($RI$–$S_w$) can be further established by using Archie's law, which can further estimate hydrate saturation [48]. They are written as the following:

$$F = \frac{R_0}{R_w} = \frac{a}{\varphi^m} \tag{7}$$

$$RI = \frac{R_t}{R_0} = \frac{b}{S_w{}^n} \tag{8}$$

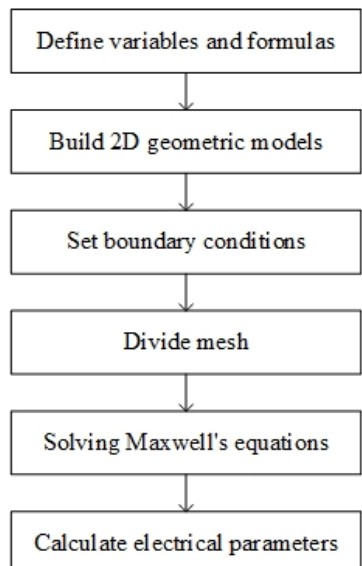

**Figure 2.** The flow chart of the numerical simulation process based on Maxwell's equations.

The formation factor $F$ is defined as the ratio of resistivity when all the pores are filled with brine, $R_0$, to the resistivity of brine, $R_w$, which is only related to the rock properties, porosity and pore structure of the formation. $\varphi$ is the effective porosity of the rock, $a$ is the lithology coefficient related to the rock properties, generally within 0.6~1.5, and $m$ is the cementation index, related to the pore structure, generally within 1.5~3. The resistivity index $RI$ is defined as the resistivity of petroliferous rocks, $R_t$, to $R_0$, which is related to water saturation and lithology. $S_w$ is the water saturation of rock, $b$ is the coefficient related to lithology, generally close to 1, and $n$ is the saturation exponent, generally close to 2.

The pore structure of the reservoir using the Archie age is simple, and the pores of the rock can be completely filled by experimental electrolytes. With the gradual deepening of oil and gas field exploration and development, the understanding of the complexity of reservoir pore structure is more in-depth, and the Archie formula needs to be improved when it is used. Therefore, using numerical simulation to calculate hydrate resistivity under different occurrence modes in the formation and constructing the relationship between resistivity and saturation is an important and effective method to study water saturation models.

## 3. Numerical Simulation of Electrical Properties at the Microscopic Pore Scale

### 3.1. Porous Medium Model

With reasonable simplification, 2D models can obtain the desired results faster and with great savings in computational resources. The equivalent substitution of 2D models for 3D models has certain limitations. Since the model simulated in this paper is homogeneous, the difference between 2D and 3D is not obvious. To facilitate research on the hydrates' electrical properties in the microscopic pore structure of the seafloor porous medium, 2D geometric models were adopted, which are equivalent to three-dimensional models while saving computational power. The marine hydrate-bearing reservoir is characterized by shallow burial, low compaction and unconsolidated cementation. Meanwhile, its porosity mainly varies from 0.2 to 0.5, with local porosity as high as 0.7 [49–52]. Based on this point, the porosity without hydrate in this paper is set to 0.6. The first step is to construct a porous medium skeleton. As shown in Figure 3, two different skeleton arrangement models were considered. Model A has a square arrangement of skeleton grains, while Model B has a 45° inclination in skeleton grain arrangement. These two models can be regarded as the skeleton arrangements formed by a more extreme stratigraphic sedimentary environment, respectively. In most cases, hydrates are present in coarse sand to medium silty sandy (500–20 μm) sediments [53–56]. Each skeleton grain has the same radius of 35.68 μm, which belongs to medium silty sandy. Further, the 2D rectangular geometry

models are the same size of 1000 × 1000 μm with 10 × 10 skeleton grains uniformly distributed. The distance between the centers of two skeleton grains $L_g$ is 100 μm. Moreover, the space around which four skeleton grains wrap is called a pore, and the space between two skeleton grains is called a throat.

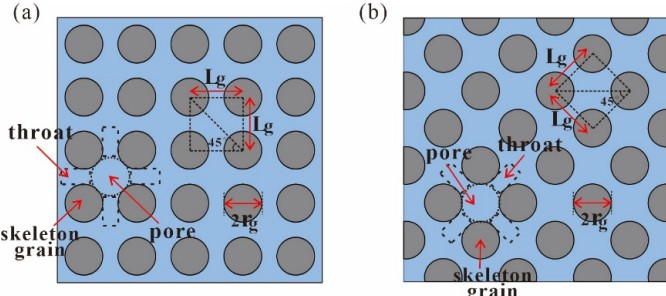

**Figure 3.** Two skeleton arrangements of porous medium models: (**a**) Model A with a square arrangement of skeleton grains; (**b**) Model B with a 45° inclination in skeleton grain arrangement.

The conductive response mechanism of hydrate can be discussed by simulating the hydrate in the pore space, including different occurrence patterns of hydrates in sediments and different distribution morphologies of hydrate in PF mode. Taking Model A as an example, Figure 4 shows the local diagram of three hydrate occurrence patterns in the model, and each model is composed of several symmetrical local diagrams. Among them, the PF mode contains five different hydrate distribution morphologies (Figure 5): circle, square, square rotated by 45° (square-45°), ellipse and ellipse rotated by 90° (ellipse-90°). It is noteworthy that each distribution morphology is suspended in the pores without contact with the skeleton grains. Finally, we constructed a series of 14 models considering multiple skeleton arrangements, occurrence patterns and distribution morphologies. In addition, various hydrate saturations can be simulated by adjusting the radius and side length of hydrate particles in each model.

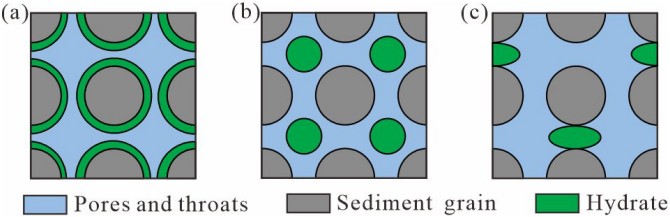

**Figure 4.** Local diagram of the hydrate occurrence patterns in the model: (**a**) grain-cementing (GC) mode; (**b**) pore-filling (PF) mode; and (**c**) load-bearing (LB) mode.

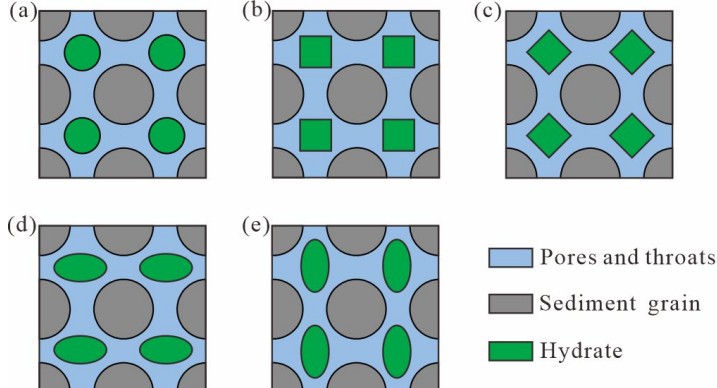

**Figure 5.** Different distribution morphologies of hydrate in PF mode: (**a**) circle; (**b**) square; (**c**) square rotated by 45° (square-45°); (**d**) ellipse; and (**e**) ellipse rotated by 90° (ellipse-90°).

### 3.2. Setting of the Numerical Simulation Model

Erickson and Jarrard (1998) found that additional conductivity of clay was not significant in highly porous marine clastic sediments and the effect of clay could be neglected [57]. In this paper, a relatively pure quartz sand medium was simulated without clay components, so the effect of its additional conductivity is also disregarded. A total of three components were set up: seawater, skeleton grain and hydrate. As it is known, seawater is highly conductive, while hydrate can be considered as completely non-conductive (i.e., electrical insulator). Conductivity is defined as expressing the strength of a substance's ability to transmit electric current, which is strongly influenced by temperature. Because the model is at the micron scale, it is assumed that the fluid inside the pores has a constant temperature. Under isothermal conditions, the effective electrical properties of the three components do not change. Therefore, their conductivity is 3 S/m, $1 \times 10^{-3}$ S/m and $1 \times 10^{-8}$ S/m, respectively [58,59].

The finite element method was used to solve the electric field model equations of the porous media based on a constant current field, and the resistivity of the porous media containing hydrate for numerical simulation was derived. The detailed procedure is described in Section 2.2. Figure 6 shows the schematic diagram of a conventional 2D porous medium under electrical simulation, which follows the law of current conservation. The top and bottom boundaries of the model are insulated, and the left and right sides are the electrical potential and ground boundaries, respectively. The left side is connected to a 0.1 A constant-current power supply, indicating current inflow, and the right side is grounded, indicating current outflow.

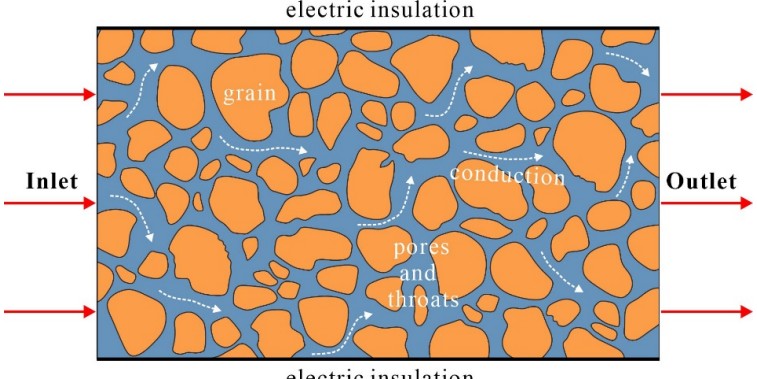

**Figure 6.** Two-dimensional diagram of electrical conduction in conventional porous media, where the white dashed line represents the direction of current conduction.

The entire computational domain including seawater, skeletal particles and hydrates was discretized using a triangular mesh that could be applied to arbitrarily complex geometries. Furthermore, too coarse mesh settings could lead to the improper resolution of thin regions and short edges. Therefore, a fine meshing method was adopted to improve accuracy during model calculation. Since the models are dominated by ion conductivity and the pores and throats are their main channels, local refinement was performed on this part of the mesh.

## 4. Results

### 4.1. Three Occurrence Patterns of Hydrates in Sediments

The resistivity of pore water (seawater) during the simulation was 0.33 $\Omega \cdot$ m, and the original core saturation resistivity was 0.78 $\Omega \cdot$ m. The effects of three hydrate occurrence patterns (PF, GC and LB mode) on resistivity were considered, as shown in Figure 7. Overall, the resistivity response of these two skeleton arrangements was generally consistent, and there was a positive correlation between resistivity and hydrate saturation in the three occurrence patterns. The curves can be roughly divided into three stages. In the first stage, when the hydrate saturation was below 0.2, the resistivity of the three modes was nearly

changeless, and the effect of hydrate occurrence patterns on electrical conduction was almost negligible in this case. In the second stage, the resistivity of the three modes of hydrate saturation changed slowly in the range of 0.2~0.4. Finally, the resistivity grew rapidly with increasing hydrate saturation, especially when it reached above 0.5. For the LB mode, even with low hydrate saturation, the resistivity was relatively high, which is more different from the other two modes. Notably, the PF mode mainly influenced the pore space, while the GC mode mainly focused on the narrow throat space. However, when the resistivity of the PF mode had a circular distribution morphology, the degree of resistivity varying with hydrate saturation was basically the same as that of the GC mode. Although the resistivity of Model B is more sensitive to the variation in hydrate saturation, the arrangement of skeleton grains has little effect on resistivity in general.

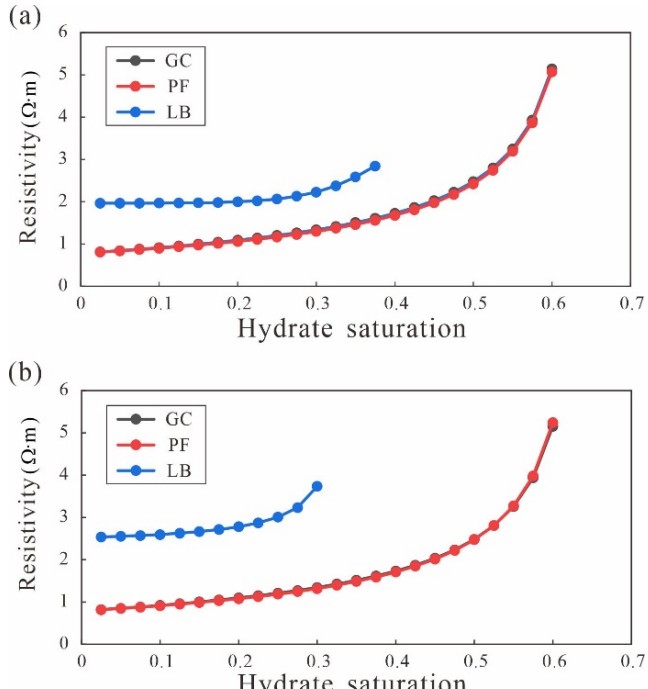

**Figure 7.** The effects of three hydrate occurrence patterns on resistivity: (**a**) Model A and (**b**) Model B.

The current density diagram allows us to visualize the distribution of the electric current during conduction. Since the simulated model was homogeneous, a partial plot was obtained for illustrative purposes. The product of electric field intensity and conductivity is the current density. During the simulation, the conductivity was assumed to be constant, and the electric field intensity and current density were proportional to each other. However, the conductivity corresponding to different components varies widely, and the current density and electric field intensity can be combined to understand the electrical properties.

The conduction path of electric current in the GC mode was basically the same as that in the non-hydrate condition, and the GC mode mainly led to a smaller throat channel but had little effect on pore space (Figure 8). With increasing hydrate saturation, the vertical gap between the two skeletal grains is the main channel of electrical conduction in Model A. Meanwhile, Model B mainly represents the gaps in each skeletal grain, which refer to the corners around one grain, and the curvature of the electrical conduction path rose. Due to the tiny conductivity of hydrate, the current density was close to zero even though the electric field intensity of hydrate becomes larger with increasing saturation. The conductivity of the seawater was constant and as the hydrate saturation increased, the electric field intensity became stronger, which is proportional to the current density.

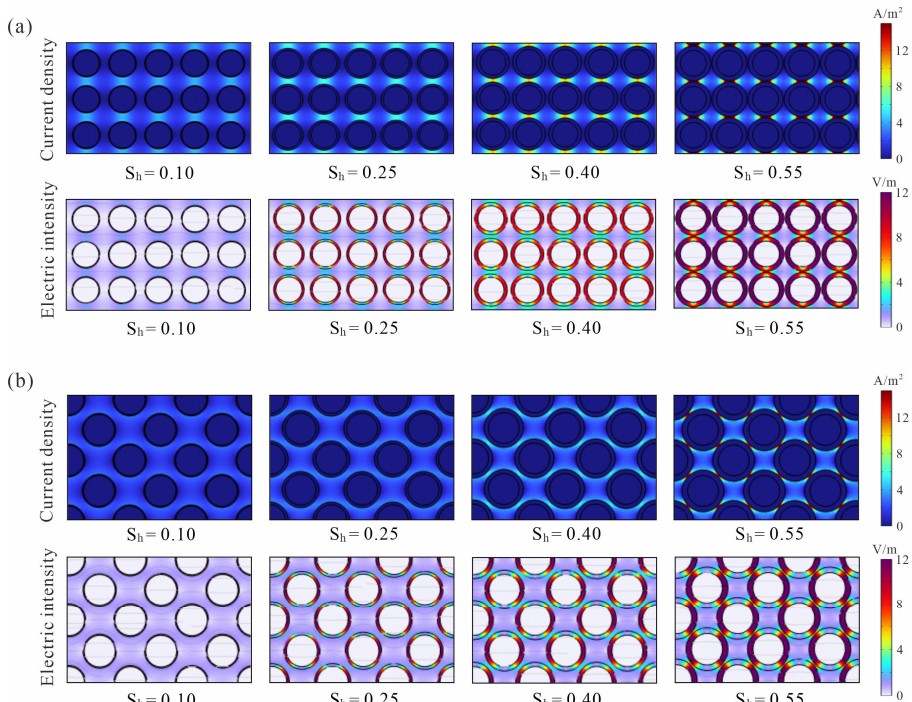

**Figure 8.** Schematic diagrams of the current conduction process in the GC mode under different sediment skeleton arrangements and hydrate saturations. Each subgraph shows the current density on top and the electric field intensity at the bottom: (**a**) Model A and (**b**) Model B.

The PF mode mainly occupied the pore space, causing it to become smaller (Figure 9). At first, the channel of electrical conduction in Model A was mainly dominated by the vertical gap of two skeletal grains, and Model B is the vertical gap between the skeletal grain and hydrate. With the increase in hydrate saturation, Model A gradually occupied the gaps in each skeleton grain as the main channel, and the curvature of the electrical conduction path grew, while the conduction path of Model B was unchanged. The electric field intensity became stronger as the hydrate saturation increased. Similar to the GC mode, the electric field intensity was also greater where the current density was high for seawater.

The hydrate in the LB mode was in contact with both skeleton grains simultaneously, and this had an impact on electrical conduction even if the saturation of the hydrate was relatively low (Figure 10). In Model A, the vertical gap between the two skeleton grains was the main channel for electrical conduction. When the hydrate saturation rose to 0.375, two hydrate grains in the lateral direction were about to come into contact and the electric current passed mainly through the narrow throat between the skeleton grains and hydrates. There was no significant change in the electrical conduction path in Model B, and the main conduction path was where the current density was high. Since the hydrate in the LB mode was contacted with the skeleton grains, the electric field intensity remained relatively strong at a low hydrate saturation, and the electric field intensity was maximal between the contact points of the hydrate and rock skeleton, which is different from the previous two modes. The pore and throat were the main channels for the ionic conductivity of hydrate-bearing sediments, and the load-bearing hydrate mainly blocked the pore and throat, which in turn caused an increase in resistivity. After the formation of sediments, the resistivity of the load-bearing hydrate was higher than in the other two modes.

In summary, hydrate saturation was low initially, and the pore space and throat channel were only a little affected by the hydrate. Meanwhile, the current density was very tiny, which was nearly uniformly distributed in the pore space, and the model's resistivity was poor. As hydrate saturation grew, the pore space and throat channel gradually became small, and the electrical conduction path was affected. All electric currents were squeezed through the narrow throat channel, causing the electric current to concentrate in the local

area. At this time, the current density at the narrow pore-throat became larger and the resistivity increased. Eventually, when the hydrate saturation increased to a certain value, the pore space was no longer connected and the saturation exponent as well as resistivity tended toward infinity.

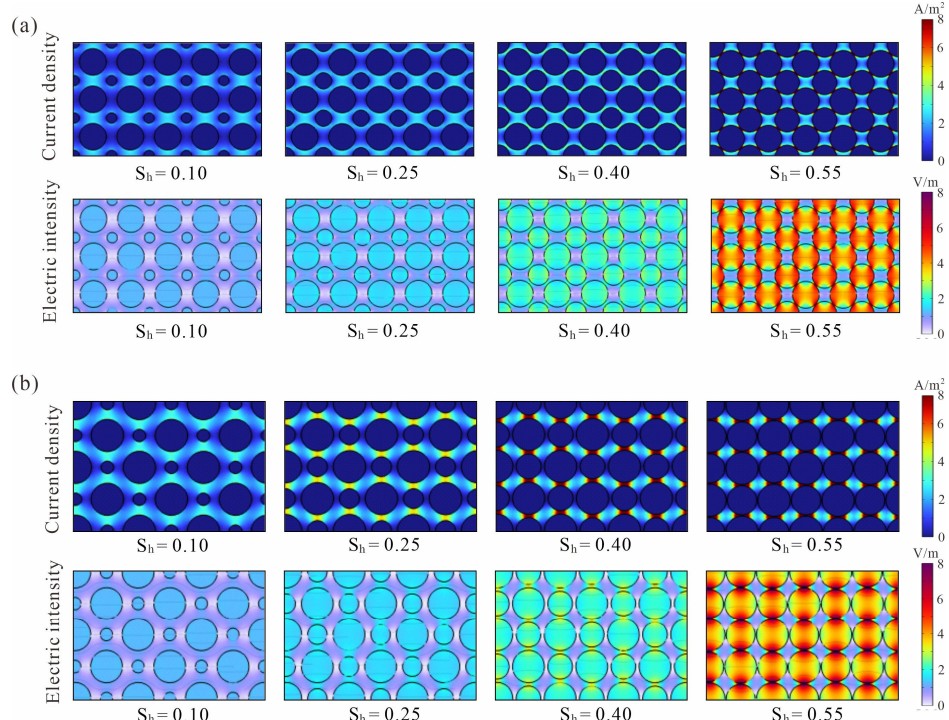

**Figure 9.** Schematic diagrams of the current conduction process in the PF mode under different sediment skeleton arrangements and hydrate saturations. Each subgraph shows the current density on top and the electric field intensity at the bottom: (**a**) Model A. (**b**) Model B.

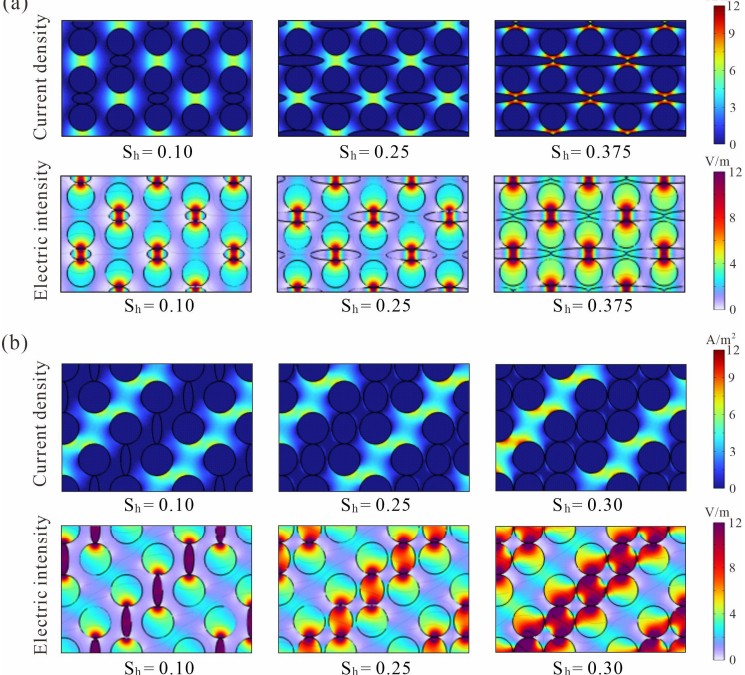

**Figure 10.** Schematic diagrams of current conduction process in the LB mode under different sediment skeleton arrangements and hydrate saturations. Each subgraph shows the current density on top and the electric field intensity at the bottom: (**a**) Model A and (**b**) Model B.

### 4.2. Five Distribution Morphologies of Hydrates in the PF Mode

The resistivity of different hydrate distribution morphologies in the PF mode was also positively correlated with hydrate saturation, but the degree of positive correlation varies considerably (Figure 11). Due to the different skeleton arrangements, the resistivity variance with hydrate saturation for both models was not exactly the same even for the distribution morphologies of hydrate in the PF mode. When hydrate saturation was less than 0.2, the resistivity of Model A essentially remained constant. Once the hydrate saturation exceeded 0.2, the resistivity began to vary obviously. The degree of resistivity variation with hydrate saturation was square, circle, ellipse, ellipse-90° and square-45° in descending order. For Model B, the hydrate saturation with 0.3 was a critical value: below 0.3, the resistivity fluctuated only a little, and above 0.3, the resistivity progressively grew with hydrate saturation. Among them, the ellipse-90° distribution morphology had a rapid rise in resistivity after the hydrate saturation was greater than 0.5. Moreover, the square and square-45° distribution morphologies had opposite forms of curve changes in the two skeleton models, owing to the fact that Model B was obtained after rotating Model A by 45°.

The electric field intensities of the PF modes varied similarly, so the electric field intensities of each distribution morphology in the PF modes are not described in this section. For the PF mode (square), Model A had less of an effect on electrical conduction when hydrate saturation was low. When hydrate saturation increased to 0.375, the four angles of hydrate mainly obstructed the electrical conduction, and the current density between the four angles and the skeleton grains became larger (Figure 12a). Model B took the vertical gap between the skeleton grains and the hydrate as the main conduction channel. As the hydrate saturation increased, the flow space became smaller and the current density at the gap rose (Figure 12b). Comparing the two models, it can be found that the current density distribution of Model A was more concentrated and the resistivity rose more quickly.

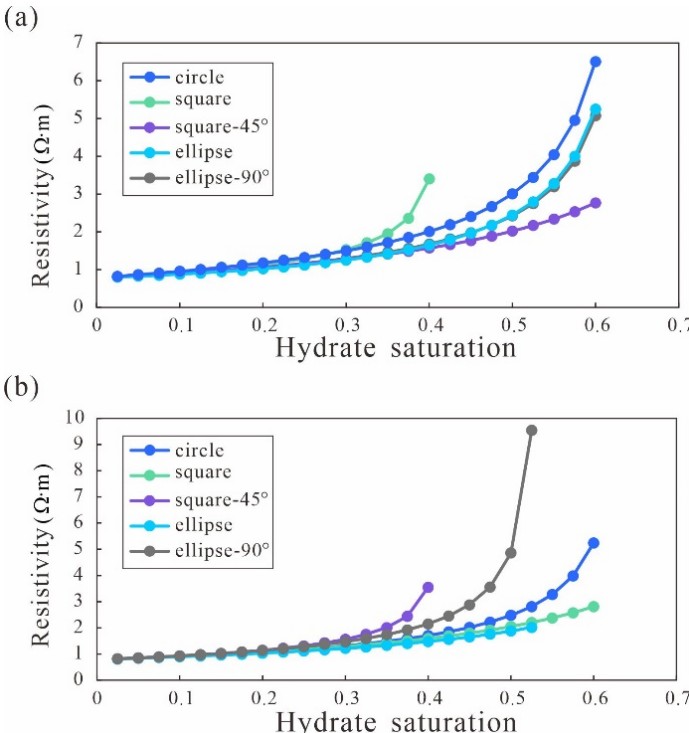

**Figure 11.** The effects of hydrate distribution morphologies in the PF mode on resistivity: (**a**) Model A and (**b**) Model B.

(a)

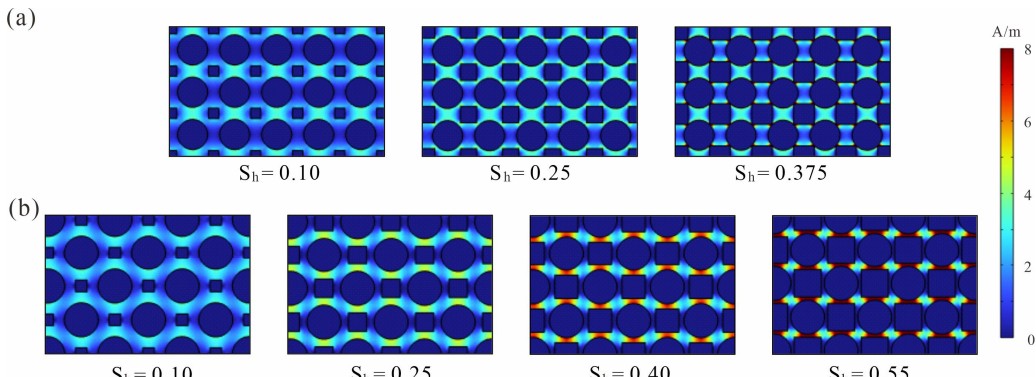

**Figure 12.** Current density diagram in the PF mode (square) with different sediment skeleton arrangements and hydrate saturations: (**a**) Model A and (**b**) Model B.

Figure 13 shows the current density distribution of the two models for the PF mode (square-45°). Model A takes the gap between two vertical skeleton grains as the main conduction channel. While at the top and bottom vertices of the hydrate, the current density becomes locally larger, indicating that the current will gather at this place to pass during the conduction process. With increasing hydrate saturation, the main conduction channel gradually changed to the gap between the hydrate and skeleton grains, as well as the top and bottom vertices of the hydrate. In Model B, the current density was more concentrated even if the hydrate saturation was 0.25. As the saturation increased, the electrical conduction channel became smaller and the electric current was more intensively passed through the throat.

(a)

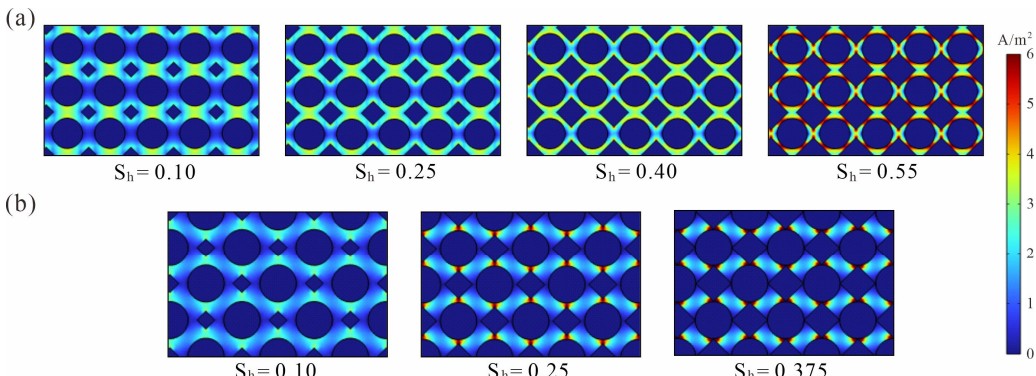

**Figure 13.** Current density diagram in the PF mode (square-45°) with different sediment skeleton arrangements and hydrate saturations: (**a**) Model A and (**b**) Model B.

Due to the difference in skeleton arrangement, Model A occupied the gap between two vertical skeleton grains as the main conduction channel, while Model B occupied the vertical gap between the skeleton grains and the hydrate. As the hydrate saturation kept growing larger, the conducting path changed and gradually became smaller, which is reflected in the concentration of current density at the local pore-throat. The current density of Model A became strong quicker at the same saturation, indicating that the PF mode (ellipse) had a greater effect on Model A (Figure 14). When the hydrate distribution morphology was ellipse-90°, the current density of Model B was already large at a hydrate saturation of 0.25 (Figure 15), showing that Model B was more sensitive to the PF mode (ellipse-90°).

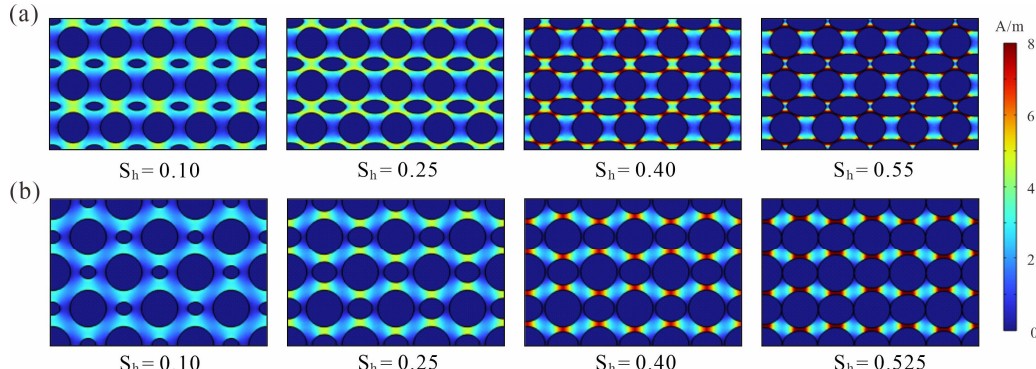

**Figure 14.** Current density diagram in the PF mode (ellipse) with different sediment skeleton arrangements and hydrate saturations: (**a**) Model A and (**b**) Model B.

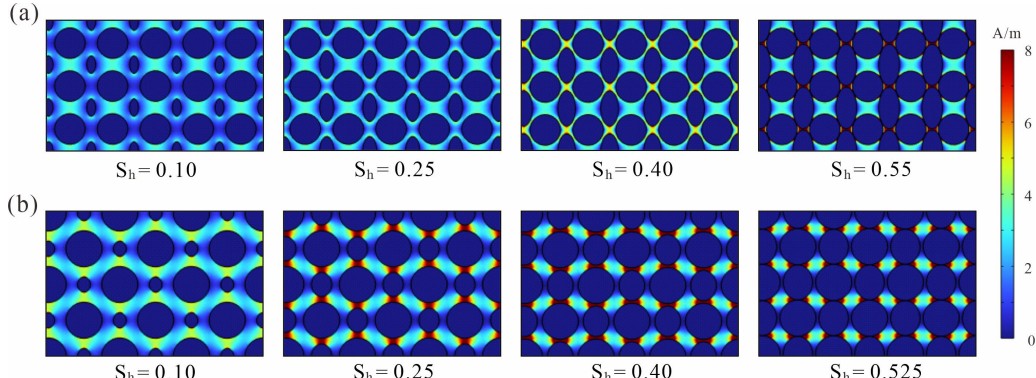

**Figure 15.** Current density diagram in the PF mode (ellipse-90°) with different sediment skeleton arrangements and hydrate saturations: (**a**) Model A and (**b**) Model B.

The above analysis shows that the skeleton arrangement and hydrate distribution morphology had different effects on the electrical conduction path. The growth of hydrates in the sediment pore space led to the reconstruction of the pore structure, and the conduction channel of the electric current became complicated, which in turn affected the electrical properties of the sediment. With the increasing hydrate saturation, the electrical conduction path changed, which in turn led to a larger resistivity. When hydrate saturation increased to a certain value, it resulted in severe blockage of electrical conduction channels and a rapid increase in current density at the pore-throat.

### 4.3. Saturation Exponent n in the Archie Formula

Typically, the resistivity of reservoir samples at several hydrocarbon saturations is measured in the laboratory to calibrate the value of $n$ in the Archie formula. Initially, Pearson measured ice-bearing sandstones to estimate the $n$ of gas hydrate and eventually obtained $n = 1.9$, which was then widely used in the hydrate industry over the following decades [60]. However, $n = 1.9$ is an empirical value and the $n$ value in a hydrate environment is related to the hydrocarbon reservoir. It depends on the sediment pore shape, pore connectivity, conductive pore water distribution, hydrate occurrence pattern and distribution morphology, etc.

The relationship between the resistivity index and water saturation ($RI$–$S_w$) was fitted to the Archie formula to obtain the empirical parameter value $n$. The log–log plots ($RI$–$S_w$) for Model A and Model B are shown in Figures 16 and 17, respectively, which do not exactly follow the Archie phenomenon. From the above analysis, it is known that the effect on resistivity is not significant when the hydrate saturation is below 0.2. If a low hydrate saturation is considered, it will lead to a poor $n$ and interfere with the prediction of hydrate saturation. Therefore, only a hydrate saturation above 0.2 was considered when fitting the

saturation exponent *n* in this paper. Both plots show that the curves of the GC mode and the PF mode (circle) basically overlap and they are in the middle of the other PF modes. Both models had large values for *n* for the PF mode (ellipse-90°), indicating that this hydrate growth mode had a large effect on electrical conduction. These modes with the largest or the smallest *n* all belonged to the PF mode. Table 1 shows the statistical *n* values for different hydrate occurrence patterns and distribution morphologies of both models. In general, the variation range of *n* for Model B was larger, demonstrating that the skeletal structure of Model B was more susceptible to the growth of hydrate compared to Model A. In other words, the resistivity of Model B was somewhat more sensitive to changes in hydrate saturation. These phenomena suggest that the arrangement of skeleton grains also has an impact on the resistivity of the hydrate-bearing reservoir.

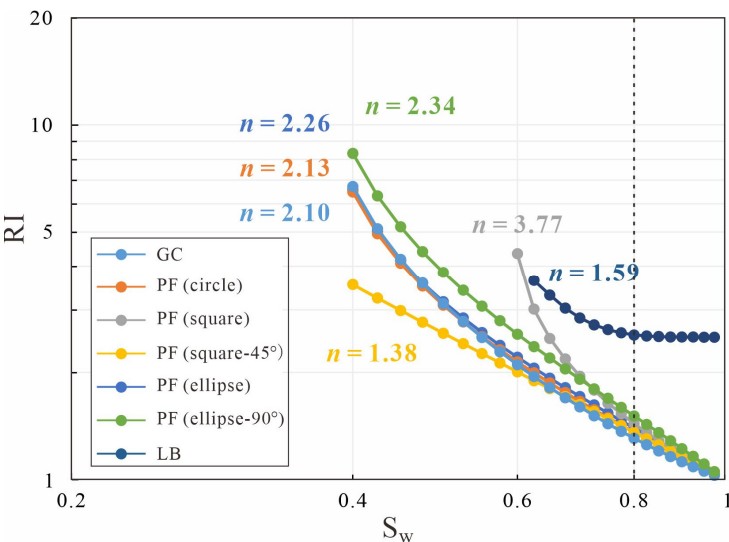

**Figure 16.** The fitting relationship between the resistivity index and water saturation in Model A.

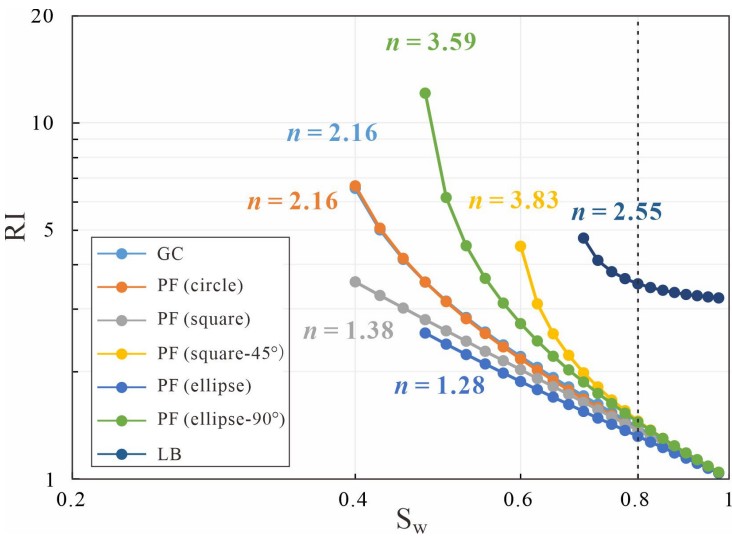

**Figure 17.** The fitting relationship between the resistivity index and water saturation in Model B.

**Table 1.** Saturation exponent *n* for different modes of two models.

| Mode | GC | PF (Circle) | PF (Square) | PF (Square-45°) | PF (Ellipse) | PF (Ellipse-90°) | LB | Mean |
|---|---|---|---|---|---|---|---|---|
| Model A | 2.10 | 2.13 | 3.77 | 1.38 | 2.26 | 2.34 | 1.59 | 2.22 |
| Model B | 2.16 | 2.16 | 1.38 | 3.83 | 1.28 | 3.59 | 2.55 | 2.42 |

Saturation exponents *n* fitted in this paper were compared with previous work using both experimental resistivity measurements of hydrate synthesis in the laboratory and actual reservoir resistivity logging data (Figure 18). The data from Ren et al. (2010) [61] and Chen et al. (2013) [62] are laboratory measurements. Mallik 5L-38 is the permafrost Mallik Gas Hydrate Production Research Well in the Northwest Territories in Canada, and the two marine wells WR313-H and GC955-H are part of the Gulf of Mexico Gas Hydrate Joint Industry Project. These three wells comprise field measured data, all from Cook and Waite (2018) [63]. The hydrate saturation of the actual reservoir resistivity logging data was greater than 0.4, and the hydrate saturation of the experimental resistivity measurement data was mainly less than 0.4. Since the in situ sediments had obvious differences from the laboratory samples in terms of porosity and degree of cementation and compaction, the saturation exponents *n* corresponding to the in situ data were higher than the experimental data. When the hydrate saturation was higher than 0.4, the trend of actual reservoir resistivity logging data was similar to that of the GC mode, and the saturation exponent was slightly larger than that of the GC mode. For high hydrate saturation ($S_h > 0.4$), the saturation exponent *n* could be taken as $2.42 \pm 0.2$, with 0.2 being the difference between the mean value of saturation exponents *n* of Model A and Model B. It can be observed that the hydrate occurrence pattern and distribution morphology were dominated by the GC and special PF modes at high hydrate saturation. In Figure 18, experimental data points are within the PF mode and outside the GC mode. When the hydrate saturation was lower than 0.4, the saturation exponent was smaller and dominated by the PF modes.

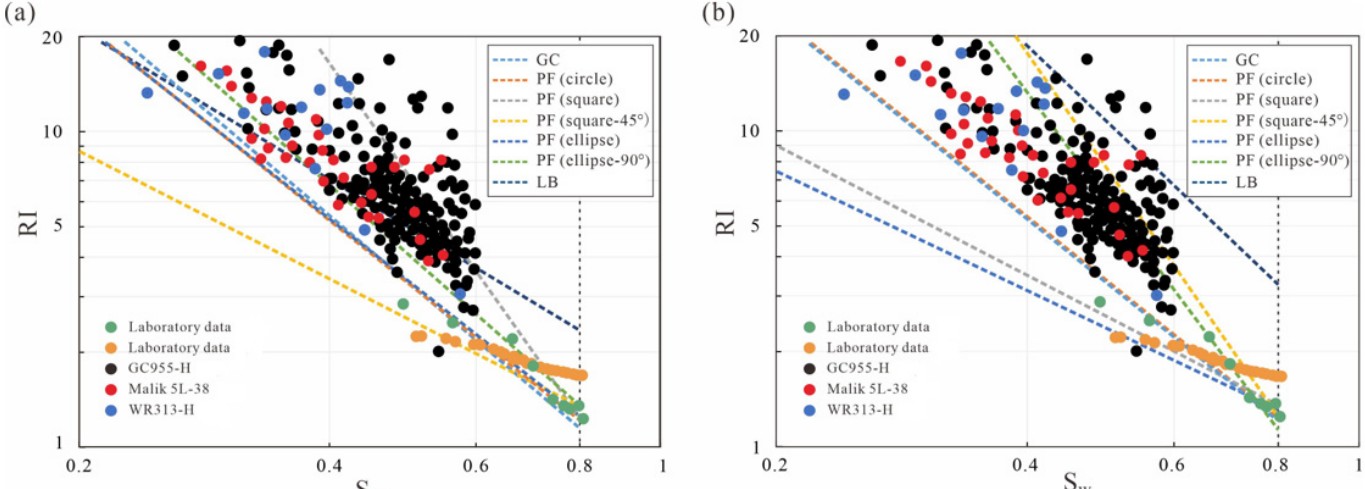

**Figure 18.** Comparison of the saturation exponents *n* fitted in this paper with previous work: (**a**) Model A and (**b**) Model B. Green dot—laboratory data [61], orange dot—laboratory data [62], black dot—GC955-H [63], red dot—Malik 5L [63] and blue dot—WR313-H [63].

### 4.4. Analyses of Influencing Factors

As solid crystalline substances, hydrates can occupy or block the electrical conduction channels of porous media, so the electrical characteristics of hydrate-bearing reservoirs are very different from those of other reservoirs. To further explore the hydrate-bearing reservoir electrical conduction mechanism, Model A was used as an example to analyze the influence of conductivity, porosity, tortuosity and the pore-throat on resistivity in this section.

In the case of high initial porosity of the reservoir, the electrical conductivity of seawater had little effect on the resistivity (Figure 19). As the initial porosity of the reservoir decreased, the ion flow space became smaller, and the electrical effect on the sediment gradually became stronger. The effects on the electrical properties of sediments were basically similar for the seawater conductivity of 2 S/m, 3 S/m and 5 S/m. When the porosity was higher than 0.5, it could be considered to have had the same degree of

influence, demonstrating that the differences between the mineralization of seawater under high porosity can be neglected in terms of the electrical properties of sediments. In contrast, the seawater conductivity of 1 S/m was significantly different from them, and this phenomenon gradually became obvious as the initial porosity of the reservoir decreased, especially when the porosity was below 0.4.

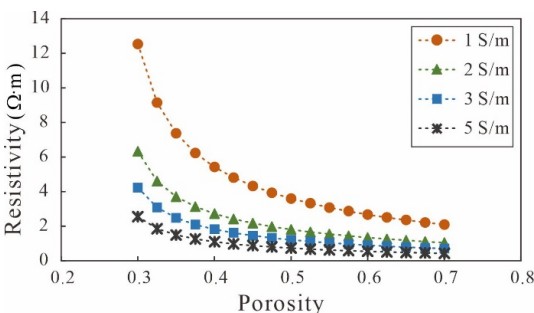

**Figure 19.** The effect of the electrical conductivity of seawater and porosity on resistivity.

The selected occurrence pattern was the PF mode (circle), and the hydrate saturation was negatively correlated with the porosity and resistivity of the sediments (Figure 20). When the hydrate saturation was 0.1 or 0.2, the resistivity was low regardless of the sediment porosity, and it can be concluded that hydrate-bearing reservoirs are hardly identified only through electrical property information at the saturation of less than 0.2. However, when the hydrate saturation was higher than 0.4, the resistivity of the sediment significantly differed from the original resistivity, i.e., there was a significant electrical logging response in the field exploration data, especially when the porosity was below 0.5. The effect of different hydrate saturation on resistivity varied greatly when porosity was low, and resistivity was similar when porosity was high. In summary, porosity had little effect on low hydrate saturation but drastically affected high hydrate saturation.

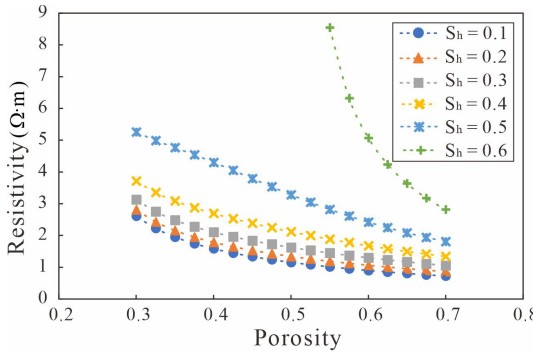

**Figure 20.** The effect of hydrate saturation and porosity on resistivity.

Since the skeleton of Model A was uniformly arranged and the tortuosity of the GC and LB modes were kept at 1.0 and 1.4, respectively, without further change, only the different distribution morphologies of hydrate in the PF mode are shown in Figure 21. With increasing hydrate saturation, the tortuosity of different distribution morphologies gradually grows and appears as obvious differentiation phenomena. In particular, the tortuosity of the PF mode (ellipse) rose most slowly with hydrate saturation, and the PF mode (square) was the fastest. As explained in Figure 10a, the order of tortuosity and resistivity ranking for different distribution morphologies at the same hydrate saturation was not completely the same. For example, the resistivity of the PF mode (square-45°) in Figure 10a changed most slowly, while the slowest change in tortuosity in Figure 20 was in the PF mode (ellipse), indicating that tortuosity was not the main factor affecting the electrical properties of the sediments.

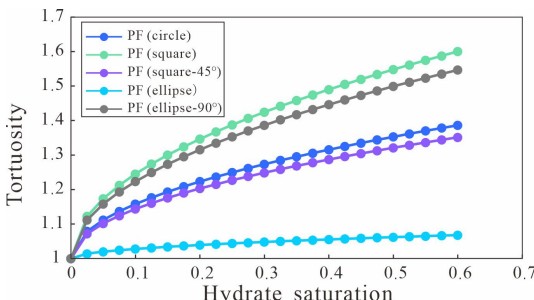

**Figure 21.** Tortuosity of five hydrate distribution morphologies with different saturations.

As we know, hydrate growth in the sediment pore space will result in the reconstruction of the pore structure, and we had to take into account the effect of pore and throat sizes on electrical properties. A new pore space and throat channel of different sizes were formed by different occurrence patterns and distribution morphologies of hydrate, as shown in Figure 22. Notably, the PF mode (ellipse) and PF mode (ellipse-90°) had the same pore and throat. Due to the LB mode's hydrate growing in the middle of the skeletal grains and occupying all the pore space of the sediment, this mode was no longer considered. The variation in the pore and throat at different hydrate saturations was statistically analyzed, and the pore space and throat channel gradually decreased with increasing hydrate saturation (Figure 23). Since the PF modes of hydrate mainly grew in the pore space, all newly generated pores were relatively small, while in the GC mode, hydrate was mainly wrapped around the outside of the skeletal grains, which had less influence on the pores. The change in the hydrate pore size in the PF mode was divided into two stages. When the hydrate saturation was small, we regarded the circle tangent of both the skeleton grains and hydrate as a pore. Further, when the hydrate saturation reached 0.4, the pore size decreased sharply, so only the tangent to the hydrate was considered and no longer the tangent to the skeleton grains. Comparison with Figure 10a showed that even though the newly generated pore size of the GC mode did not vary strongly with hydrate saturation, the resistivity was relatively sensitive to hydrate saturation, which means that the pore size affected the electrical conduction mechanism but was not its main factor. The influence of hydrate saturation on the throat was relatively significant, and the size of the throat gradually tended toward zero with the increase in saturation, where the PF (square) mode had the greatest influence on the throat. The variation in throat size with hydrate saturation was basically the same as resistivity with hydrate saturation in Figure 23, which indicates that throat size is one of the most essential parameters affecting the electrical characteristics of hydrate-bearing reservoirs.

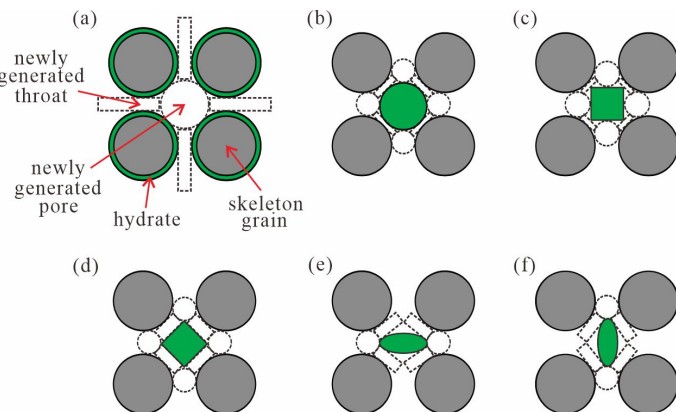

**Figure 22.** The newly generated pores and throats formed by different occurrence patterns and distribution morphologies of hydrate: (**a**) GC mode; (**b**) PF (circle) mode; (**c**) PF (square) mode; (**d**) PF (square-45°) mode; (**e**) PF (ellipse) mode and (**f**) PF (ellipse-90°) mode.

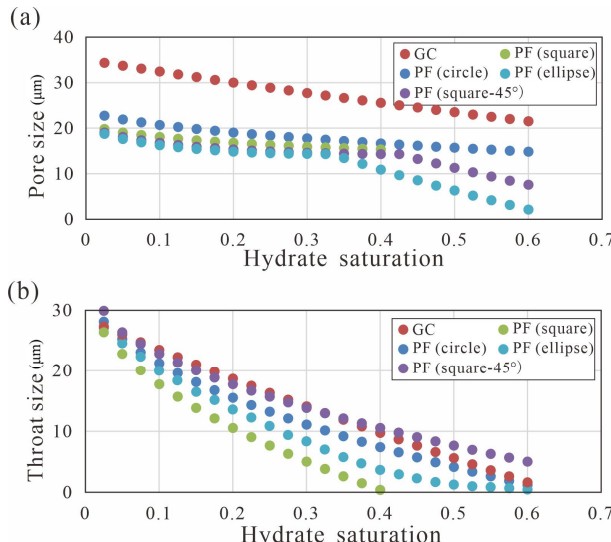

**Figure 23.** Sizes of newly generated pores and throats with different hydrate saturation: (**a**) Pore size and (**b**) Throat size.

## 5. Discussion and Limitations

To facilitate the study of the electrical properties of hydrate in different occurrence patterns and distribution morphologies, homogeneous sediments were simulated but the influence of free gas on the electrical property relationship was not considered in this paper. Since the model simulated in this paper is homogeneous, the difference between the 2D and 3D models is not obvious. Meanwhile, we mainly simulated hydrate-bearing sediments of medium fine sandstone. Due to the high salinity of seawater, hydrates in the sea are predominantly ionically conductive, and pores and throats are their main conduction channels. Actual marine sediment is heterogeneous and contains many impurities, which mainly affect the distribution of pores and throats. The complex pore structure significantly impacts the electrical properties of the sediment, and it is necessary to study electrical laws based on the actual composition of seabed sediment. Currently, digital core technology is an important means to study this factor. The lower part of the hydrate reservoir was close to a gas source, and there was a large amount of free gas, which can be regarded as an insulator. The migration and distribution of free gas may have hindered the migration of charged particles and caused an increase in resistivity. Meanwhile, once the temperature and pressure conditions changed, hydrate also dissociated and changed from a solid to gas state. Therefore, future work could focus more on the study of the electrical properties of hydrate reservoirs under the co-existence of solid–liquid–gas phases, as well as complex pore structures.

## 6. Conclusions

Based on the study of marine hydrate-bearing sediments, constant current field theory was used for simulation to construct numerical samples of hydrate with different occurrence patterns and distribution morphologies, and the electrical conduction mechanism in the pore scale of hydrate-bearing sediments was investigated. The following conclusions were obtained:

(1) The changes in resistivity can be roughly divided into three stages for multiple occurrence patterns and distribution morphologies. In the first stage, when the hydrate saturation was below 0.2, the effect on electrical conduction was almost negligible. In the second stage, resistivity grew slowly in the range of a hydrate saturation of 0.2~0.4. Finally, resistivity increased rapidly with hydrate saturation when the hydrate saturation reached above 0.5.

(2) For the LB mode, resistivity was relatively high even when the hydrate saturation was low, which is quite different from the other two modes. The resistivity of different distribution morphologies of hydrate in the PF mode varied greatly with hydrate saturation, and even with the same hydrate distribution morphology, the resistivity changes in the two models were not completely the same.

(3) The saturation exponent *n*, which was obtained by the fitting relationship between the resistivity index and water saturation, varied widely for different distribution morphologies of the PF mode. After a comparison with previous work, it is suggested that for sediments with a high hydrate saturation ($S_h > 0.4$), the saturation exponent *n* can be taken as $2.42 \pm 0.2$.

(4) The skeleton grain arrangement of the sediment, initial porosity, hydrate saturation, conductivity of seawater, tortuosity, pore and throat size all complicate the conductive mechanism of electric current, which in turn affects the electrical properties of hydrate-bearing sediments, with the size of the throat being the most critical factor affecting resistivity.

**Author Contributions:** Research design, X.L. and C.W.; Investigation, X.L. and S.W.; simulations, X.L., C.W. and Y.Z.; data analysis, X.L., C.Z. and C.P.; writing—original draft preparation, X.L.; supervision, C.Z. and C.P.; writing—review and editing, C.P. All authors have read and agreed to the published version of the manuscript.

**Funding:** This work is supported by the National Natural Science Foundation of China [No. 42274232].

**Institutional Review Board Statement:** Not applicable.

**Informed Consent Statement:** Not applicable.

**Data Availability Statement:** The datasets generated during the current study are available from the corresponding author upon reasonable request.

**Conflicts of Interest:** The authors declare no conflict of interest.

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
