# Peer review of "Numerical Simulation of Electrical Properties for Pore-Scale Hydrate-Bearing Sediments with Different Occurrence Patterns and Distribution Morphologies"

_jmse, doi:10.3390/jmse11061169_

Round 1
Reviewer 1 Report
Authors present a fundamentally theoretical work, valid and robust in its treatment, but do not present any verification from an experimental point of view.
Four remarks:
1) Line 456. "For high hydrate saturation (Sh > 0.4) the exponent n can be taken as 2.42+/- 0.2" This is correct, but if Sh <0.4 the models A and B don' t work? Why?
2) I don' t understand why if Sh < 0.4 is dominant the PF model?
3) Is no verification possible from an experimental point of view utilizing real experimental data?
4) Lines 155-171, The general physic discussion is too elementary and can be omitted.
Scientific English used in the work is clear.
Author Response
Dear Reviewer:
Thank you for your comments concerning our manuscript entitled “Numerical simulation of electrical properties for pore-scale hydrate-bearing sediments with different occurrence patterns and distribution morphologies”. Those comments are all valuable and very helpful for revising and improving our manuscript, as well as the important guiding significance to our researches. We have studied comments carefully and have made corrections. The main corrections in the manuscript and the responds to comments are as following:
Please see the attachment.

Reviewer 2 Report
This study explores the effect of hydrate pore-scale habit on the resistivity of the host sediment. The authors simulate the presence of hydrate crystals at different pore-scale habits, including pore-filling, load-bearing, and grain-cementing, and obtain the corresponding resistivities. The work is interesting, novel, and within the aim and scope of JMSE and I believe it can be published after the authors carefully address the following comments:
1. Generally, the abstract and main body of the manuscript needs to be proofread as I see some minor English typos and grammatical errors. Please also make sure the text is well-punctuated.
2. Can the authors add a couple of sentences about the simulation method they used in this work? This would help the reader to obtain a better view of the work.
3. The Introduction is interesting and reads well but is inadequate and could be enriched.
- Can I ask the authors to be more specific and provide more details about the potential hazards associated with the dissociation of natural gas hydrates? For instance, “… and other disasters …” is not very clear to the reader.
- Can the authors add a brief discussion regarding how the co-presence of hydrates and the other phases, particularly ice, can change the physical properties of the host sediment? They may refer to the following articles:
o Dong et al., 2019. Developing a new hydrate saturation calculation model for hydrate-bearing sediments. Fuel, 248, pp.27-37.
o Wu et al., 2020. Pore‐scale 3D morphological modeling and physical characterization of hydrate‐bearing sediment based on computed tomography. Journal of Geophysical Research: Solid Earth, 125(12), p.e2020JB020570.
o Farahani et al., 2021. Insights into the climate-driven evolution of gas hydrate-bearing permafrost sediments: Implications for prediction of environmental impacts and security of energy in cold regions. RSC advances, 11(24), pp.14334-14346.
o Farahani et al., 2021. Development of a coupled geophysical–geothermal scheme for quantification of hydrates in gas hydrate-bearing permafrost sediments. Physical Chemistry Chemical Physics, 23(42), pp.24249-24264.
o Zhao et al., 2022. Pore-Scale Investigation of the Electrical Property and Saturation Exponent of Archie’s Law in Hydrate-Bearing Sediments. Journal of Marine Science and Engineering, 10(1), p.111.
- About “ … the anomalous change of resistivity …”, can the authors discuss if there are any other sources of the anomalous resistivity behaviour, or if the only driver is the presence of hydrates?
- Can the authors add a brief and preferably critical literature review of the recent studies conducted by scholars for the determination of electrical properties of hydrate-bearing sediments?
4. In section 2.1, I would refer the authors to the following articles where different pore-scale habits of hydrates occurrence in coarse and fine porous media have been reviewed:
o Ren et al., 2020. Permeability of hydrate-bearing sediments. Earth-Science Reviews, 202, p.103100.
5. Can the authors provide a schematic diagram regarding the procedure followed in section 2 to obtain the electrical resistivity?
6. “ … which are equivalent to the three-dimensional models while saving computational power.” The authors developed their model in 2D. Can they comment on what they would expect if the model were developed in 3D? Several transport phenomena in porous media such as fluid flow and heat transfer can be very different, depending upon the dimension.
7. I think the authors consider the porous media fully saturated with water prior to placing the hydrates crystal in the pore space with different habits. When hydrate is present, the pore space is essentially filled with water and hydrates. Would this assumption be very simplifying? It is given that the process of hydrate formation in porous media requires one another phase to be present, i.e. natural gas, and the presence of natural gas can highly influence the resistivity.
8. The authors consider homogeneous porous structures. Can they comment on how the results will be affected if heterogeneity, as a very important factor, comes to play?
9. The authors mention they use FEM to solve the electrical field model and obtain the resistivity. How do they achieve this? Do they use a specific package, or do they develop the code by themselves? More information about the solution process is essential to be added to the manuscript.
10. I believe it would be a good idea if the authors clarify that they do not consider the effect of clay presence in their current porous media model.
11. What references have been used to take the intrinsic resistivity values from? Can the authors add relevant references?
12. Can the authors discuss why the resistivities obtained from the load-bearing cases are generally higher than those obtained from the other habits? Could this be related to the arrangement of the load-bearing hydrate crystal and the current direction?
13. As a general suggestion, I would ask the authors to enrich the caption of the figures, particularly figures 7-9 and 11-14 as these figures are not self-explanatory.
14. Can I ask the authors to add a paragraph, detailing the implications and propose the room for improvement of this work?
Generally, the abstract and main body of the manuscript needs to be proofread as I see some minor English typos and grammatical errors. Please also make sure the text is well-punctuated.
Author Response

(The authors gave the same response as above.)

Round 2
Reviewer 1 Report
Paper may be published, all integrations are fitting and congruent.
Minor editing of English are required.
Author Response
Dear Reviewer:
Thank you for your comments and recognition!
Reviewer 2 Report
I evaluated the authors' responses to my comments and the revised manuscript. I see some improvements however not being convinced about their response to the following comments:
Comment#2) The authors should add a couple of sentences in the abstract and the introduction about their simulation method.
Comment#3) I like the discussion provided regarding the co-existence of hydrates and the other components and their effect on the electrical properties. I strongly believe they should bring this into their manuscript and refer the reader to some relevant articles including those suggested.
Comment#4) I see no action regarding this comment in the manuscript. I believe the authors should expand their discussion to let the reader know if they deal will coarse or fine sand.
Comment#6) Again, I see no action regarding this comment in the manuscript. It would be great if they can add some discussions.
Comment#12) The same concern, not any action. Please add your detailed discussion to the manuscript.
I believe the manuscript will be publishable after the above comments are properly addressed by the authors.
Author Response
Dear Reviewer:
Thank you for your comments. We have studied comments carefully and have made corrections. The main corrections in the manuscript and the responds to comments are as following:
